# Activation of PKCε-ALDH2 Axis Prevents 4-HNE-Induced Pain in Mice

**DOI:** 10.3390/biom11121798

**Published:** 2021-11-30

**Authors:** Bárbara B. Martins, Natália G. Hösch, Queren A. Alcantara, Grant R. Budas, Che-Hong Chen, Daria Mochly-Rosen, Julio C. B. Ferreira, Vanessa O. Zambelli

**Affiliations:** 1Laboratory of Pain and Signaling, Butantan Institute, São Paulo 05503-900, SP, Brazil; barbara.martins@esib.butantan.gov.br (B.B.M.); natalia.hosh@esib.butantan.gov.br (N.G.H.); queren.alcantara@esib.butantan.gov.br (Q.A.A.); 2Gilead Sciences Inc., Foster City, CA 94404, USA; grantbudas@gmail.com; 3Department of Chemical and Systems Biology, School of Medicine, Stanford University, Stanford, CA 94305, USA; chehong@stanford.edu (C.-H.C.); mochly@stanford.edu (D.M.-R.); 4Department of Anatomy, Institute of Biomedical Sciences, University of Sao Paulo, Sao Paulo 05508-090, SP, Brazil; jcesarbf@usp.br

**Keywords:** oxidative stress, neuroprotection, protein kinase, neurodegeneration, hyperalgesia

## Abstract

Protein kinase Cε (PKCε) is highly expressed in nociceptor neurons and its activation has been reported as pro-nociceptive. Intriguingly, we previously demonstrated that activation of the mitochondrial PKCε substrate aldehyde dehydrogenase-2 (ALDH2) results in anti-nociceptive effects. ALDH2 is a major enzyme responsible for the clearance of 4-hydroxy-2-nonenal (4-HNE), an oxidative stress byproduct accumulated in inflammatory conditions and sufficient to induce pain hypersensitivity in rodents. Here we determined the contribution of the PKCε-ALDH2 axis during 4-HNE-induced mechanical hypersensitivity. Using knockout mice, we demonstrated that PKCε is essential for the nociception recovery during 4-HNE-induced hypersensitivity. We also found that ALDH2 deficient knockin mice display increased 4-HNE-induced nociceptive behavior. As proof of concept, the use of a selective peptide activator of PKCε (ΨεHSP90), which favors PKCε translocation to mitochondria and activation of PKCε-ALDH2 axis, was sufficient to block 4-HNE-induced hypersensitivity in WT, but not in ALDH2-deficient mice. Similarly, ΨεHSP90 administration prevented mechanical hypersensitivity induced by endogenous production of 4-HNE after carrageenan injection. These findings provide evidence that selective activation of mitochondrial PKCε-ALDH2 axis is important to mitigate aldehyde-mediated pain in rodents, suggesting that ΨεHSP90 and small molecules that mimic it may be a potential treatment for patients with pain.

## 1. Introduction

Pain is a common and disabling complication of inflammatory and neuropathic condition that affects over 1 billion people worldwide [1]. Its related costs increase annually, reaching around US$ 620 billion/year in the United States alone [2]. The treatment of pain involves the use of therapies that mainly cause symptom relief and are limited by their low efficacy and/or numerous adverse effects. In this scenario, a major coordinated effort towards identification and validation of promising targets affecting pain is critical for further screening and development of more effective and safe interventions.

Protein kinase Cε (PKCε) is a serine and threonine kinase that is highly expressed in sensory neurons and when activated it translocates to multiple intracellular sites [3]. PKCε activation contributes to hypernociception by increasing the membrane ionic current [4,5,6]. Other studies demonstrated that the PKCε antagonist (εV1-2) induces analgesia in multiple pre-clinical models of inflammatory and neuropathic pain [7,8]. However, despite promising preclinical results, clinical trials for acute severe postoperative pain and chronic postherpetic neuralgia failed to show a beneficial effect of εV1-2 as an analgesic [7]. These negative findings can be explained by the fact that, regardless of its upstream signaling, the same protein kinase can activate different signaling pathways simultaneously (both essential and detrimental) and therefore affect the effectiveness and safety of PKC inhibitors in the long-term. Under this scenario, PKCε modulators that selectively favors only essential-kinase substrate interaction or inhibits detrimental-kinase substrate interaction might have better efficacy to treat pain.

Emerging evidence suggests that PKCε promotes cell survival by regulating mitochondrial function and signaling. In this regard, we previously described mitochondrial aldehyde dehydrogenase 2 (ALDH2) as one of the PKCε substrate, whose activity inversely correlates with hypersensitivity in mice [9]. ALDH2 plays a critical role in metabolizing reactive aldehydes, accumulating during ethanol consumption (i.e., acetaldehyde) and oxidative stress (i.e., 4-hydroxy-2-nonenal, 4-HNE), and it regulates pain-like behavior in multiple models in rodents [10]. Moreover, knock-in ALDH2*2 mice with decreased ALDH2 activity display exacerbated 4-HNE accumulation and pain-like behaviors [9,10]. However, the ALDH2 contribution to aldehydes-induced pain is still unknown.

4-HNE is an endogenous α,β-unsaturated aldehyde generated during oxidative stress, mainly through lipid peroxidation in the mitochondria. Increased 4-HNE adducts in the injured site positively correlates with pain [11]. We previously demonstrated that accumulation of 4-HNE plays a major role in the establishment and progression of pain [9]. Elevated 4-HNE levels were also found in carrageenan-induced inflammation [9], endometriosis [12], spinal cord injury (SCI) [13], experimental autoimmune encephalomyelitis (EAE) [14] and migraine [15]. Mechanistically, 4-HNE can form protein adducts with cysteine, histidine, and lysine residues in proteins via Michael addition; therefore, causing inhibition of protein function and generating pain [16]. For example, 4-HNE targets a transient receptor potential ankyrin 1 (TRPA1), which is highly expressed in sensory nerve fibers and drives pain signaling [17].

Considering the negative effects of 4-HNE accumulation in inflammation and pain, it is expected that increasing 4-HNE clearance through selective activation of PKCε-ALDH2 axis will positively affect pain outcome. Therefore, using gain and loss of function strategies, we set out to determine the role of PKCε-ALDH2 axis in 4-HNE-induced nociception in mice. We also tested here the possibility that selective pharmacological activation of mitochondrial PKCε-ALDH2 axis counteracts aldehydic load and improves pain-related behaviors triggered by 4-HNE.

## 2. Materials and Methods

### 2.1. Animals

Experiments were approved by the Ethics Committee on the Use of Experimental Animals at the Butantan Institute (São Paulo, Brazil) and were performed according to the Committee for Research and Ethical Issues of the International Association for the Study of Pain (IASP). In this study we used male mice weighing 20–22 g. The following strains were used: wild-type mice C57BL/6 mice; heterozygous for PKCε (PKCε^+/−^) as previously described [18]; transgenic mice carrying the Asian ALDH2*2 alleles that carry the E487K mutation in the ALDH2 enzyme [9]. They were housed in an appropriate room, with water and food *ad libitum* with soundproofing, controlled temperature (22 ± 1 °C), and light-dark cycle (12/12 h).

### 2.2. Reagents

4-Hydroxy-2-nonenal (4-HNE, Sigma, St. Louis, MO, USA), HC-030031 (TRPA1 antagonist, I-TRPA1, Sigma, Barueri, SP, Brazil), Alda-1 (Acme Chemical, Mountain View, CA, USA), Indomethacin (Sigma, St. Louis, MO, USA), TAT (American Peptide Co., Sunnyvale, CA, USA), εV1-2 (American Peptide Co.), ΨεRack (Ontores Peptide, Hangzhou, China), ΨεHSP90 (Ontores Peptide).

### 2.3. Drug Administration

4-HNE was administered on the plantar surface in different doses (15–120 nmol per paw in 20 µL) or its vehicle (1 nM HCL in saline) [16]. I-TRPA1 (HC 030031; 300 nmol in 20 µL) or its vehicle (10% DMSO, 5% Tween 80 and 85% saline) was administered on the plantar surface [19,20]. Alda-1 was administered subcutaneously (10 mg/kg) and DMSO and PEG (1:1) were its vehicle [9]. Indomethacin was administered intraperitoneally (10 mg/kg) and 5% sodium bicarbonate in PBS was its vehicle [21]. All vehicles described above are represented as control. For the peptides, the control group was the TAT (YGRKKRRQRRR) carrier administered via intraplantar (1 μg in 20 µL of saline). εV1-2 (EAVSLKPT) was administered via intraplantar (1 μg in 20 µL of saline) 30 min before 4-HNE and intrathecal (1.35 μg in 10 μL of saline) 15 min before 4-HNE [22,23,24]. ΨεRack (HDAPIGY) was administered via intraplantar (1 μg in 20 µL of saline) [6]. ψεHSP90 (PKDNEER; 1 μg in 20 µL of saline) was administered via intraplantar [25]. All peptides were synthesized at Ontores. At displayed purity of 95–99%.

### 2.4. Behavioral Testing

In vivo behavioral testing was blindly performed between 7:00 AM and 1:00 PM in an isolated, temperature-, and light-controlled room. Animals were placed on a mesh floor inside a red acrylic box individually, suspended approximately 30 cm from the bench, allowing access to the paws. Mice were habituated to the testing apparatus for 30 min on the day before and the day of the test. The filament was inserted below the mesh floor and onto the plantar skin until the filament just bent. The method used was “up and down” [26]. Thus, a von Frey 9-filament logarithmic series (Aesthesiometer Semmer-Weinstein monofilaments, Stoelting Co., Wood Dale, IL, USA) was employed. This method uses stimulus oscillation around the response threshold to determine the median 50% threshold of the response. The tests started with the 0.6 g filament. The von Frey filaments were applied six times and the paw removal or not was recorded.

### 2.5. Western Blot

The samples were homogenized in a lysis buffer containing Hepes-NaOH (1 M, PH 7.9), NaCl (1.54 M), EGTA (200 mM), Triton-X 100 (1%), and protease and phosphatase inhibitors (1:300). Polyacrylamide gel electrophoresis (SDS-PAGE 10% or 8%) was used in a mini gel device (Mini-Protean, Biorad, Santo Amaro, SP, Brazil). After separation by electrophoresis, the proteins were transferred to nitrocellulose membranes (BioRad). The membranes were blocked in TBST (20 mM Tris-HCL, 150 mM NaCl, and 0.1% Tween 20) containing 5% BSA for 2 h, followed by incubation with primary antibody against HNE reduced Michael Adducts (1:1000, Millipore, Burlington, MA, USA), ALDH2 (1:500; Abcam, Waltham, MA, USA) or PKCε (1:500; Santa Cruz Biotechnology, Dallas, TX, USA) overnight, at 4 °C. Subsequently, the membranes were then incubated, for 2 h at room temperature, in the appropriate peroxidase-conjugated secondary antibody (1:5000; anti-rabbit IgG or anti-goat IgG, Sigma, USA) and developed using enhanced chemiluminescence. Quantification analysis was performed using the UVITEC software (UVITEC Cambridge, Cambridge, UK). Densitometry data were normalized to GAPDH (1:5000; Abcam) and represented as %.

### 2.6. Statistical Analysis

The statistical analysis was generated using the GraphPad Prism 8 program (GraphPad Software Inc., San Diego, CA, USA). Data are expressed as mean ± SEM. Statistical evaluation of the data was conducted using two-way analysis of variance (ANOVA), followed by Tukey’s post hoc test. Statistical *t*-student analysis was performed to analyze Figure 2B,F,H. The significance index considered was *p* < 0.05.

## 3. Results

### 3.1. Disruption of PKCε-ALDH2 Axis Contributes to 4-HNE-Induced Mechanical Hypersensitivity

Reactive oxygen species-induced peroxidation of polyunsaturated fatty acids produces 4-HNE, which contributes to pain-like behavior in rodents [27]. To characterize the direct effect of 4-HNE on the mechanical nociceptive threshold, we administered 4-HNE in different doses (15–120 nmol/paw) (Figure 1A). Our results showed that the doses of 15 and 30 nmol/paw does not affect the mechanical threshold. However, 60 and 120 nmol/paw of 4-HNE induce mechanical hypersensitivity, reaching the peak 30 min after 4-HNE administration (Figure 1B). Therefore, we selected the 60 nmol/paw dose for the time course study. The 4-HNE hypersensitivity effect persists for 8 h (Figure 1C).

Next, to characterize the role of PKCε in 4-HNE-induced mechanical hypersensitivity, we used a genetic mouse model with decreased PKCε expression (Figure 2A). As expected, PKCε protein levels were significantly reduced in plantar tissue from heterozygous PKCε^+/−^ mice when compared with WT littermates (Figure 2B). Interestingly, PKCε^+/−^ mice showed extended 4-HNE-induced mechanical hypersensitivity when compared with wild-type (Figure 2C,D). These data suggest that PKCε contributes to 4-HNE-induced nociception recovery in mice.

To further characterize the role PKCε-ALDH2 axis in 4-HNE-induced nociception, we next used knockin mice carrying the inactivating Lys487 point mutation in ALDH2, identical to the mutation found in Han Chinese [28]; denoted ALDH2*2 [9]. We evaluated the effect of 4-HNE injection in the hind paw of ALDH2*2 and WT mice (Figure 2E). As expected, ALDH2 protein level was reduced in the paw of ALDH2*2 mice when compared with WT (Figure 2F). The behavioral experiment showed that ALDH2*2 animals display persistent hypersensitivity that lasted 24 h (Figure 2G). Moreover, consistent with increased hypersensitivity 24 h after 4-HNE administration, ALDH2*2 mice present higher 4-HNE adduct levels in their paws when compared with WT (Figure 2H). Overall, our results using genetically modified animals provide evidence that both PKCε and ALDH2 are independent and critical players in counteracting 4-HNE-induced nociception in mice. Next, we tested whether selective pharmacological activation of mitochondrial PKCε-ALDH2 axis counteracts aldehydic load and improves pain-related behaviors triggered by 4-HNE.

### 3.2. Selective Activation of PKCε in Mitochondria Blocks 4-HNE-Induced Hypersensitivity

We previously reported that acute administration of ΨεHSP90, a peptide that increases mitochondrial PKCε translocation, induces cardioprotection by favoring PKCε phosphorylation/activation of ALDH2 [29,30] (Figure 3A). Here, we used a similar approach to test whether ΨεHSP90-induced mitochondrial PKCε translocation, and consequent activation of PKCε-ALDH2 axis, prevents 4-HNE-induced hypersensitivity in mice (Figure 3B). First, we replicated the literature findings showing that ΨεRACK-induced global activation of PKCε is sufficient to induce hypersensitivity in mice (Figure 3C). Of interest, the administration of ΨεHSP90 per se did not cause pain in mice (Figure 3C). As expected, ΨεHSP90 did not neutralize hypersensitivity induced by the global PKCε activator, ΨεRACK [6,31], when co-administered (Figure 3C).

Next, we tested the efficacy of ΨεHSP90 peptide in reducing 4-HNE-induced hypersensitivity. Our findings demonstrated that intraplantar administration of ΨεHSP90 peptide induces a sustained improvement of nociceptive threshold in mice injected with 4-HNE (Figure 3D). However, the pretreatment with the global activator (ΨεRACK) does not prevent 4-HNE-induced hypersensitivity (Figure 3D). To further prove that PKCε-induced analgesia works through ALDH2 activation, we treated ALDH2*2 deficient mice with ΨεHSP90 before 4-HNE administration (Figure 3D). Our results showed that genetically modified mice with impaired ALDH2 activity are resistant to ΨεHSP90-induced benefits during exogenous 4-HNE-induced hypersensitivity (Figure 3E). These findings suggest that functional ALDH2 is required to mediate PKCε-induced analgesia under aldehydic load conditions.

To further characterize mitochondrial ALDH2 as a downstream PKCε effector, we next used a selective small molecule activator of ALDH2, termed Alda-1 [29], during 4-HNE-induced nociception (Figure 4A). Of interest, administration of Alda-1 before the insult was sufficient to completely prevent 4-HNE-induced hypersensitivity in mice (Figure 4B); therefore, validating the critical role of ALDH2 in counteracting aldehyde-induced pain. Indeed, we provided evidence that direct activation of ALDH2 can circumvent the requirement of PKCε to reduce pain.

### 3.3. Selective Activation of PKCε in Mitochondria Prevents Carrageenan-Induced Hypersensitivity

Next, we asked whether activation of mitochondrial PKCε-ALDH2 axis blocks mechanical hypersensitivity induced by endogenous production of 4-HNE. Our group previously demonstrated that 4-HNE accumulates in the hind paw and becomes a key mediator in carrageenan-induced mechanical hypersensitivity [9]. To explore the benefits of activating mitochondrial PKCε-ALDH2 axis during carrageenan-mediated inflammatory pain, we treated WT mice with ΨεHSP90 before carrageenan injection (Figure 4C). As shown in the Figure 4D, ΨεHSP90 blocked carrageenan-induced inflammatory nociception, providing evidence that PKCε activation of ALDH2 is sufficient to prevent inflammatory pain-like behavior.

Mechanistically, 4-HNE triggers nociception, at least in part, by activating TRPA1 receptors [16]. As expected, I-TRPA1 (HC 030031), a selective TRPA1 receptor antagonist, prevented 4-HNE-induced hypersensitivity (Figure 5B). Importantly, the hypersensitive effect of 4-HNE was not a consequence of an inflammatory response, since indomethacin, a non-steroidal anti-inflammatory drug, did not prevent the 4-HNE-induced hypersensitivity (Figure 5C).

Finally, some groups claim that PKCε is highly expressed in sensory neurons and plays an important role in triggering nociception by activating TRPV1 receptors [32,33] and/or Nav1.8 channels [6] in the plasmatic membrane. In fact, PKCε inhibitor peptide (εV1-2) prevents hypersensitivity in several rodent models of pain [22,31,34]. However, whether pharmacological PKCε inhibition affects 4-HNE-induced hypersensitivity was not determined. To address this question, we assessed the role of peripheral and central PKCε inhibition in 4-HNE-induced nociception. For that, we injected εV1-2 intraplantarly or intrathecally prior to 4-HNE administration. Our results demonstrated that neither peripheral nor central PKCε inhibition affects 4-HNE-induced mechanical hypersensitivity during the first 2 h after 4-HNE injection in mice (Figure 5E,F), similar to our data observed in PKCε KO mice.

## 4. Discussion

In this study, using strategies of loss and gain of function, we showed that the PKCε-ALDH2 axis plays an important role in 4-HNE-induced nociception. Using transgenic mice, we showed that a reduction in PKCε or ALDH2 levels strongly compromises the 4-HNE removal and prolongs the nociception. Importantly, selectively activating mitochondrial PKCε (with ΨεHSP90) or ALDH2 (with Alda-1) successfully prevented 4-HNE-induced hypersensitivity. Finally, we showed that the treatment with ΨεHSP90 is sufficient to prevent carrageenan-induced nociception, a model in which endogenous 4-HNE accumulates and became a key mediator of nociception. Our study provided evidence that the design of molecules that competitively disrupt a specific protein–protein interaction (kinase-chaperone), such as, ΨεHSP90, is a promising approach to develop more feasible drugs that selectively affect only detrimental kinase substrate interactions in pain pathophysiology.

Aldehydes are highly diffusible and reactive agents generated during numerous painful processes. 4-HNE is a relatively stable compound that can travel remarkable distances from the site of synthesis. This aldehyde can reach measurable concentrations in the tissues and in the blood. Its physiological concentration is in the sub micromolar range (<0.1 μM), while in oxidative stress, even micromolar levels can be observed [35,36]. Therefore, the concentration of 4-HNE presently used may represent levels compatible with 4-HNE-produced in a pathological condition. 4-HNE has been implicated in tissue damage, dysfunction, and pain associated with neurodegenerative diseases such as multiple sclerosis, cancer, diabetes, endometriosis, among others [11,37]. The mechanisms by which 4-HNE contributes to these pathologies are unknown, however, it is well established that the sustained lipid peroxidation and further accumulation of 4-HNE lead to impaired mitochondrial function, which further contribute to the disease progress. Therefore, understanding the molecular pathways involved in 4-HNE control may be of great importance for treating multiple pain related pathologies.

We showed previously that ALDH2 is a PKCε substrate [10,29] and a regulator of acute nociception in rodents. Therefore, we tested the hypothesis that the disruption in PKCε-ALDH2 axis would aggravate 4-HNE-induced nociception. Our data showed that 4-HNE induces extended hypernociception in both PKCε and ALDH2*2 mice (Figure 2C,G). This effect was followed by 4-HNE accumulation in paws of ALDH2*2 mice, confirming that ALDH2 is essential for 4-HNE detoxification. We have previously showed that ALDH2*2 are more sensitive to carrageenan, acetaldehyde and formalin-induced nociceptive behavior than wild-type mice [9]. However, whether PKCε^+/−^ mice also accumulate 4-HNE as a consequence of ALDH2 impairment remains to be investigated.

Previous studies demonstrated that full knockout PKCε mice show decreased epinephrine-induced hyperalgesia [5]. However, to our knowledge, there are no studies evaluating endogenous aldehydes-mediated nociception, for example, induced by carrageenan, in full knockout PKCε mice. Therefore, further studies are necessary to clarify the role of PKCε-ALDH2 axis in those models. A limitation of our study is our inability to breed full knockout (homozygotic) mice in our laboratory. However, heterozygous may have advantages since the remaining protein expression lowers the chance of unexpected compensatory mechanisms activation [38].

Mitochondrial translocation of PKCε is mediated by HSP90, which permits mitochondrial import of PKCε by the import receptor Tom20 [39]. ΨεHSP90 is a 7-amino acid peptide activator of PKCε, derived from an HSP90 homologous sequence located in the C2 domain of PKCε. Importantly, ΨεHSP90 activates ALDH2 without inducing PKCε translocation to the plasma membrane [30]. This was confirmed by our behavior data showing that, differently from the global activator ΨεRACK, ΨεHSP90 does not induce hypernociception when injected alone. Then, we asked whether ΨεHSP90 would have a beneficial effect in 4-HNE-induced mechanical hypersensitivity. Interestingly, our results demonstrated that this peptide significantly prevents 4-HNE-induced mechanical hypersensitivity. Furthermore, to prove that PKCε-induced analgesia works through ALDH2 activation, we showed that ΨεHSP90 did not prevent the mechanical hypersensitivity in ALDH2*2 mice. Because ALDH2*2 variant induces a structural deficit in both the coenzyme-binding and active sites of ALDH2 [40], ΨεHSP90 failed to rescue ALDH2 activity, losing its analgesic effect. These data confirm again that PKCε-induced analgesia requires a functional ALDH2 to remove toxic aldehydes. Importantly, we also showed that the ALDH2 activation with Alda-1 is sufficient to prevent 4-HNE-induced nociception. This is in agreement with the previously report that Alda-1 administration induces analgesia in multiple pain models, associated with a significant reduction of 4-HNE adducts [9,12,13]. Therefore, with this study we confirm that 4-HNE clearance is sufficient to induce analgesia even by exogenous administration.

Aiming to translate the ΨεHSP90 analgesic effect into a broadly inflammatory pain model, we tested the peptide in carrageenan-induced mechanical hypersensitivity. ΨεHSP90 was effective in preventing inflammatory pain-like behaviors, confirming our hypothesis that PKCε activation may have a protective effect by controlling the mitochondrial ALDH2 activity. As mentioned before, the PKCε global activation in sensory neurons is pro-nociceptive by depolarizing and/or sensitizing afferent neurons following inflammatory insult [41,42,43]. Here, we showed, for the first time, that PKCε may have a protective effect through the ALDH2 axis. We also showed that the εV1-2 peptide (global PKCε antagonist) did not interfere with 4-HNE mechanical hypersensitivity when administered either peripherally or intrathecally. Importantly, εV1-2 blocks PKCε translocation and reduces hyperalgesia in a variety of inflammatory and neuropathic pain rodent models [22,31,34]. However, considering that PKCε activates different signaling pathways simultaneously, the global PKCε inhibition may compromise essential downstream pathways, such as, ALDH2-induced 4-HNE removal from the injured tissue.

Finally, we demonstrated that TRPA1 activation mediates 4-HNE-induced mechanical hypersensitivity. Trevisani and colleagues [16] have demonstrated that the dose of 4-HNE presently used, 60 nmol per paw, induces spontaneous nocifensive behavior in mice, such as paw licking and lifting, immediately after administration and this effect is mediated by TRPA1 receptors. Our study demonstrated that 60 nmol of 4-HNE lowers the mechanical threshold for >8 h and that Alda-1 prevented 4-HNE-induced mechanical hypersensitivity. Therefore, it is very likely that besides activating TRPA-1 receptors, the exogenous 4-HNE triggers endogenous 4-HNE formation, which is prevented by ALDH2 activation with Alda-1. Further studies are necessary to investigate this hypothesis; for example, Schwann cells expressing TRPA1 receptors recruit macrophages upon activation, triggering oxidative stress and 4-HNE formation [44]. Therefore, it is possible that Alda-1 would fail to block 4-HNE-induced nociception immediately after the aldehyde injection, for example 5–20 min afterwards, the time point that 4-HNE may directly induce nociception by activating TRPA1 receptors. Also, 4-HNE itself may inactivate ALDH2, thus limiting its own removal. Therefore, the endogenous 4-HNE release possibly contributes to the pain-like behavior detected in longer time points such as 4–8 h in the wild type mice and 24 h in the ALDH2*2 mice.

In our study, we did not investigate which are the main cells expressing PKCε-ALDH2 in the paw tissue, however, mitochondria are widely transported along axons and dendrites to the regions of higher energy demands, such as injured tissue. However, although neurons express ALDH2, there is recent evidence showing that spinal astrocytic ALDH2, but not neuronal, mediates alcohol-induced analgesia via acetate-GABA metabolic pathway [45]. Therefore, the ALDH2 functions go beyond aldehyde detoxification in the pathophysiology of pain.

Regardless of the signaling pathways involved in PKCε activity in the nociceptors, designing therapeutics limiting reactive aldehyde production may provide treatments particularly helpful as a pre-treatment when pain is expected, such as post-surgeries.

## 5. Conclusions

In conclusion, our results demonstrate that ALDH2 is an important PKCε substrate in attenuating 4-HNE hypersensitive effects. Targeting mitochondrial PKCε may become a good strategy to mitigate inflammatory pain.

## Figures and Tables

**Figure 1 biomolecules-11-01798-f001:**
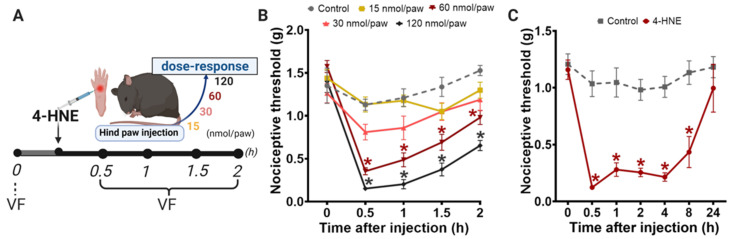
Dose-response curve and time course of 4-HNE-induced mechanical hypersensitivity. (**A**) Schematic panel of pain threshold assessment by von Frey filament test (VF) and the treatment protocol. (**B**) 4-HNE was injected intraplantarly (15, 30, 60, and 120 nmol/paw) and the nociceptive threshold was assessed by von Frey filaments, by an “up-down” method. (**C**) Time-course of 4-HNE-induced mechanical hypersensitivity (60 nmol/paw). Error bars represent mean ± SEM; *n* = 8 per condition. * *p* < 0.05 when compared with the baseline measure (time 0). Two-way analyses of variance (ANOVA) with post-hoc testing by Tukey. The observer was blinded to the experimental conditions.

**Figure 2 biomolecules-11-01798-f002:**
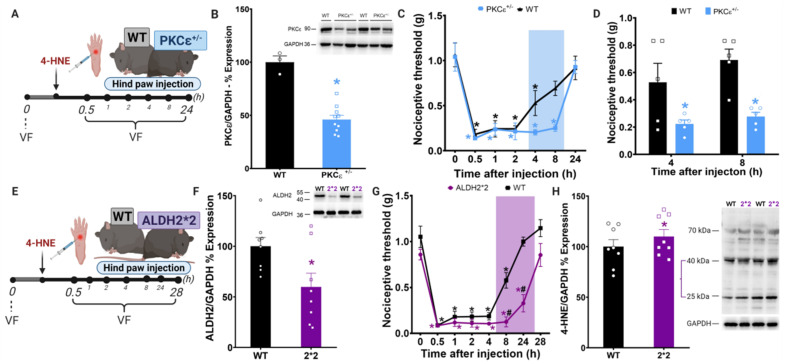
Disruption of PKCε-ALDH2 axis contributes to 4-HNE-induced mechanical hypersensitivity. (**A**) Schematic panel of drug administration and von Frey filament test (VF) in PKCε ^+/−^ animals. (**B**) PKCε expression in the paw tissue of wild-type and PKCε^+/−^ mice. T-test * *p* <0.05 compared to wild type. *n* = 8/group. (**C**) Time-course of 4-HNE-induced mechanical hypersensitivity (60 nmol/paw) in PKCε^+/−^. Two-way analysis of variance (ANOVA) with post-hoc testing by Tukey * *p* < 0.05 compared to baseline. *n* = 8/group. (**D**) PKCε^+/−^ animals have more hypersensitivity when compared to wild-type animals at 4 and 8 h after 4-HNE injection. One-way analysis of variance (ANOVA) with post-hoc testing by Tukey. * *p* < 0.05 compared to wild-type. *n* = 8/group. (**E**) Schematic panel of drug administration and von Frey filament test in ALDH2*2 animals (mice with impairment in ALDH2 activity). *n* = 8/group. (**F**) ALDH2 expression in paw tissues of wild type and ALDH2*2. *T*-test * *p* < 0.05 compared to wild type. *n* = 8/group. (**G**) Impairment in ALDH2 activity (ALDH2*2) increases the hypersensitivity when compared to wild-type animals at 8 and 24 h after 4-HNE injection. Two-way analysis of variance (ANOVA) with post-hoc testing by Tukey * *p* < 0.05 compared to baseline, # *p* < 0.05 compared to wild-type. *n* = 8/group. (**H**) representative western blot and % change in 4-HNE protein adducts in proteins extracted from paws treated with 4-HNE (60 nmol/paw) of wild type and ALDH2*2 mice (GAPDH used as loading control). *T*-test * *p* < 0.05 compared to wild type. *n* = 8/group. All data are means ± SEM. Pain assessment was carried out by an observer blinded to the experimental conditions.

**Figure 3 biomolecules-11-01798-f003:**
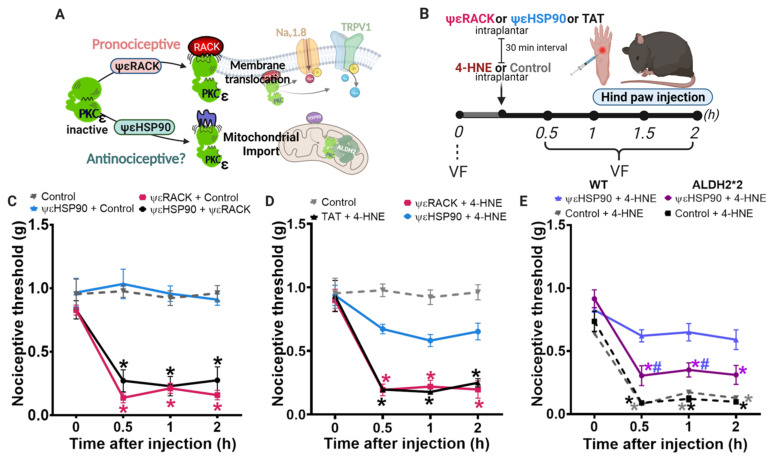
The global activation of PKCε is pro-nociceptive and the mitochondrial activation of PKCε blocks 4-HNE-induced nociception. (**A**) Schematic panel showing cytosolic and mitochondrial targets of PKCε. (**B**) Schematic panel of drug administration and von Frey filament test (VF) in wild type or ALDH2*2 animals (mice with impairment in ALDH2 activity). (**C**) ΨεHSP90 (1 μg/paw), ΨεRACK (1 μg/paw) i.pl. were administered 30 min before the vehicle or ΨεHSP90 (black line). (**D**) ΨεHSP90 (1 μg/paw) or TAT (1 μg/paw) were administered 30 min before the 4-HNE or vehicle (gray line). (**E**) WT and ALDH2*2 received ΨεHSP90 or TAT (1 μg/paw). Two-way analyses of variance (ANOVA) with post-hoc testing by Tukey. * *p* <0.05 when compared with the baseline (time 0). # *p* < 0.05 when compared to the vehicle group (control). Error bars represent mean ± SEM; *n* = 6–8 per condition. Pain assessment was carried out by an observer blinded to the experimental conditions.

**Figure 4 biomolecules-11-01798-f004:**
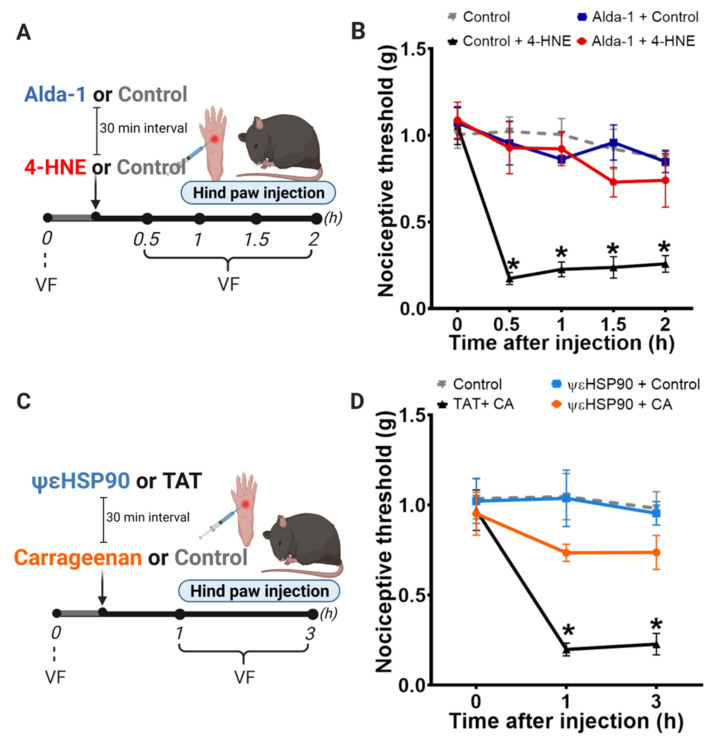
Activation of ALDH2 by Alda-1 prevents 4-HNE-induced nociception and by ΨεHSP90 prevents carrageenan-induced nociception. (**A**) Schematic panel of drug administration and von Frey filament (VF). (**B**) Wild type animals received Alda-1 (10 mg/kg) s.c. 30 min before 4-HNE (60 nmol/paw) i.pl. (**C**) Schematic panel of drug administration and von Frey filament (VF) (**D**) Wild type animals received ΨεHSP90 or TAT (1 μg/paw), i.pl., 30 min before carrageenan (CA) (100 μg/paw)/paw) or saline i.pl. Error bars represent mean ± SEM; *n* = 8 per condition. Two-way analyses of variance (ANOVA) with post-hoc testing by Tukey. * *p* < 0.05 when compared with the baseline (time 0). Pain assessment was carried out by an observer blinded to the experimental conditions.

**Figure 5 biomolecules-11-01798-f005:**
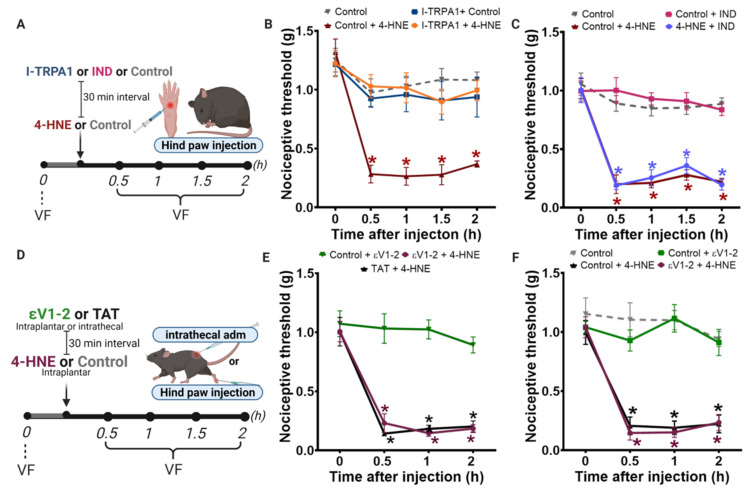
TRPA1 is involved in 4-HNE-induced nociception and the global PKCε antagonist does not prevent this hypersensitivity effect. (**A**) Schematic panel of drug administration and von Frey filament (VF). (**B**) I-TRPA1 (HC 030031; 300 nmol/paw) or vehicle were administered 30 min before 4-HNE (60 nmol/paw). (**C**) Indomethacin (10 mg/kg, i.p) or vehicle (i.p) were administered 30 min before 4-HNE (60 nmol/paw). (**D**) Schematic panel of drug administration and von Frey filament (VF). (**E**) εV1-2 (1 µg/paw) i.pl. was administered 30 min before the injection of 4-HNE (60 nmol/paw) i.pl. (**F**) εV1-2 (1.35 µg/animal) i.t. was administered 15 min before 4-HNE (60 nmol/paw). Error bars represent mean ± SEM; *n* = 8 per condition. Two-way analyses of variance (ANOVA) with post-hoc testing by Tukey. * *p* < 0.05 when compared with the baseline (time 0). Pain assessment was carried out by an observer blinded to the experimental conditions.

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
