# Peer review of "Activation of PKCε-ALDH2 Axis Prevents 4-HNE-Induced Pain in Mice"

_biomolecules, 2021, doi:10.3390/biom11121798_

Round 1
Reviewer 1 Report
Using 4-hydroxy-2-nonenal (4-HNE) inducd acute pain mouse model, the authors described the effect ofProtein kinase Cε (PKCε) in relation to the aldehyde dehydrogenase-2 (ALDH2).
The authors showed 4-HNE is the acute pain inducer in a dose dependent manner and 4-HNE induced mechannical allodynia was attenuated in PKCε +/-mouse and ALDH2*2 mice.
The main outcomes of the experiments are from the mouse behavial test using von Frey fibers. The expreriments were well-designed and the data are convincing.
While this is an interesting and a solid study, the reviewer concerns that the conclusion is not fully supported by the data. All the behavial experiments are acute model examining around 2-days. It is important, however, it seems to much to generalize that PKCε-ALDH2 axis is critical to mitigate aldehyde-mediated pain
Author Response
Comments and Suggestions for Authors #1:
Using 4-hydroxy-2-nonenal (4-HNE) induced acute pain mouse model, the authors described the effect of Protein kinase Cε (PKCε) in relation to the aldehyde dehydrogenase-2 (ALDH2). The authors showed 4-HNE is the acute pain inducer in a dose dependent manner and 4-HNE induced mechanical allodynia was attenuated in PKCε +/-mouse and ALDH2*2 mice.
The main outcomes of the experiments are from the mouse behavioral test using von Frey fibers. The experiments were well-designed and the data are convincing.
While this is an interesting and a solid study, the reviewer concerns that the conclusion is not fully supported by the data. All the behavioral experiments are acute model examining around 2-days. It is important, however, it seems to much to generalize that PKCε-ALDH2 axis is critical to mitigate aldehyde-mediated pain
Response: We would like to thank the reviewer for the suggestions and the positive evaluation of our work. Based on your comment, we would like to kindly point out that indeed we showed that 4-HNE is the acute pain inducer in a dose-dependent manner, however, 4-HNE induced mechanical allodynia was increased in PKCε +/- and ALDH2*2 mice.
As for the reviewer's concerns about the conclusion, we agreed and modified the text as follows: “In conclusion, our results demonstrate that ALDH2 is an important PKCε substrate in attenuating 4-HNE hypersensitive effects”.
Reviewer 2 Report
This is a well-written manuscript. It is interesting to read. The results are clearly presented with meaningful schematic panels. I felt that it would be better if histology analysis including PKCε, 4-HNE, and c-fos level/distribution could be included.
Author Response
Comments and Suggestions for Authors #2:
This is a well-written manuscript. It is interesting to read. The results are clearly presented with meaningful schematic panels. I felt that it would be better if histology analysis including PKCε, 4-HNE, and c-fos level/distribution could be included.
Response: We would like to thank the referee for the positive evaluation and suggestions.
We agree that histology is informative, however, the evaluation of PKCε activation requires separating membrane, mitochondrial and/or cytosolic fractions. Because we are using the mouse paw and/or DRG samples, this procedure becomes technically challenging. Unfortunately, a simple immunohistochemistry for PKCε would not give us the mechanistic information to strengthen our data. In terms of localization, studies demonstrate that PKCε is expressed in nociceptors, including the peripheral terminals (J Pharmacol Exp Ther 2004;309(2):616-25; Neuron Volume 23, Issue 3, July 1999, Pages 617-624), highly expressed in the brain, but rather scarce in the spinal cord (ENSG00000171132 code at the Protein Atlas (https://www.proteinatlas.org/) and TS Omics platforms (https://tsomics.shinyapps.io/RNA_vs_protein/)). Further studies are necessary to clarify the contribution of each cell in our model.
In the present manuscript, the 4-HNE levels were detected by western blot. This is the gold standard method for detecting the Michael adducts, allowing high sensitivity detection and accurate quantification of 4-HNE (Free Radic Biol Med 2006;41(12):1847-59; Journal of Molecular and Cellular Cardiology 2014; (71): 92-104; Sci Transl Med 2014;(251):25). Immunostaining of 4-HNE has been rather diffusive over cell compartments and cell types, as shown by De Logu and collaborators (J Clin Invest. 2019, Fig.5b). Therefore it would not be as much informative in terms of localization in the paw tissue.
Regarding c-Fos levels/distribution analysis it would clarify which neurons/regions are activated by 4-HNE and, we agree that it would be helpful to explore the 4-HNE effects in the central nervous system. Although this is an interesting exploratory study, c-Fos analysis was not at the scope of our study, but we will consider it in the following experiments.

Reviewer 3 Report
This work aims to identify a PKC subunit-ALDH2 axis in the treatment of 4-HNE-induced pain. This research direction is potentially interesting and important for the mechanism of action for ALDH2-mediated analgesic effect at the spinal and DRG level. Although important, a few major concerns can be raised to question the main hypothesis of this paper.
Major comments
- It seems the evidence to support the title of this manuscript is very thin. Most of the experimental results from PKC KO and ALDH2KI mice and pharmacological manipulations are completely parallel. The only data supporting PCK-ALDH2 axis is presented in Fig. 3E. Unless more new experimental evidence can be provided, the current title has not been fully supported by the data presented in this study.
- It is likely mouse ALDH2 is primarily expressed in mitochondria, however, there is evidence that human ALDH2 is expressed in either cytoplasmic and mitochondrial pools. It is therefore not very accurate to make a title as mitochondrial axis. To keep the current tile, the authors should carry out in vitro experiments to reveal the direct interaction between PKCe and ALDH2 in cultured astrocytes.
- 3E needs to show the control using ALDH2 KI mice given 4HNE.
- In text, there is no description of Fig. 4C.
- There are many statements throughout the entire manuscript that are rather speculative and oversold. For instance, in the Text, 256-257, “These findings strengthen our hypothesis that selective activation of mitochondrial PKCe-ALDH2 axis……”. This statement appears even before Fig. 3E.
Manor comments
- What is the cell -types of PKCe-ALDH2 axis in paw tissues?
- There is new evidence for astrocytic ALDH2 in the mechanism of action for analgesia. These recent studies should be cited and discussed.
- Is DRG and spinal PKCe-ALDH2 also involved in suppression of 4-HNE-induced pain?
- Is this PCKe-ALDH2 axis specific for in producing analgesia against 4-HNE-induced pain?
Author Response
Comments and Suggestions for Authors #3:
This work aims to identify a PKC subunit-ALDH2 axis in the treatment of 4-HNE-induced pain. This research direction is potentially interesting and important for the mechanism of action for ALDH2-mediated analgesic effect at the spinal and DRG level. Although important, a few major concerns can be raised to question the main hypothesis of this paper.
Major comments
It seems the evidence to support the title of this manuscript is very thin. Most of the experimental results from PKC KO and ALDH2KI mice and pharmacological manipulations are completely parallel. The only data supporting the PCK-ALDH2 axis is presented in Fig. 3E. Unless more new experimental evidence can be provided, the current title has not been fully supported by the data presented in this study.
It is likely mouse ALDH2 is primarily expressed in mitochondria, however, there is evidence that human ALDH2 is expressed in either cytoplasmic and mitochondrial pools. It is therefore not very accurate to make a title as mitochondrial axis. To keep the current tile, the authors should carry out in vitro experiments to reveal the direct interaction between PKCe and ALDH2 in cultured astrocytes.
Response: We would like to thank the reviewer for the suggestions and comments. The word mitochondria was removed from the title and the new title is: “Activation of PKCε-ALDH2 axis prevents 4-HNE-induced pain in mice”
3E needs to show the control using ALDH2 KI mice given 4HNE.
Response: We followed the reviewer's instruction and performed a new experiment including controls for the 4-HNE in both ALDH2 KO and WT.
In text, there is no description of Fig. 4C.
Response: The description was included.
There are many statements throughout the entire manuscript that are rather speculative and oversold. For instance, in the Text, 256-257
, “These findings strengthen our hypothesis that selective activation of mitochondrial PKCe-ALDH2 axis……”. This statement appears even before Fig. 3E.
Response: This statement was removed from the manuscript.
Manor comments
What is the cell -types of PKCe-ALDH2 axis in paw tissues?
Response: According to the best of our knowledge, there is no study showing which cells express PKCε-ALDH2 in the paw tissue. However, mitochondria are transported along axons and dendrites to the regions of higher energy demands (Cereb Cortex 2018 Oct 1;28(10):3673-3684.), such as injured tissue and PKCε is highly expressed in the DRG (Neuron Volume 23, Issue 3, July 1999, Pages 617-624; J Pharmacol Exp Ther 2004;309(2):616-25), meaning that PKCε-ALDH2 may be concentrated at the peripheral terminals. Also, there is evidence showing that macrophages are involved in the release of 4-HNE, which activates the TRPA1 channel in Schwann cells. This interaction is important to drive the hypersensitivity detected in different models of pain (De Logu F., et al. Brain Behav Immun. (2020); De Logu, F., et al. Nat Commun (2017), and De Logu F., et al. J Clin Invest. (2019)). Considering recent evidence showing that specialized cutaneous Schwann cells initiate pain sensation (Science 2019;365(6454):695-699) we speculate that peripheral terminals/Schwann and macrophages are the cells involved in the PKCε-ALDH2 axis. Of course, this hypothesis remains to be investigated.
There is new evidence for astrocytic ALDH2 in the mechanism of action for analgesia. These recent studies should be cited and discussed.
Response: Thank you for the suggestion. These studies are now cited and discussed: “In our study, we did not investigate which are the main cells expressing PKCε-ALDH2 in the paw tissue, however, mitochondria are widely transported along axons and dendrites to the regions of higher energy demands, such as injured tissue. However, although neurons express ALDH2, there is recent evidence showing that spinal astrocytic ALDH2, but not neuronal, mediates alcohol-induced analgesia via acetate-GABA metabolic pathway. Therefore, the ALDH2 functions go beyond aldehyde detoxification in the pathophysiology of pain.”
Is DRG and spinal PKCe-ALDH2 also involved in suppression of 4-HNE-induced pain?
Response: PKCε is highly expressed in DRG and less expressed in spinal cord (https://www.proteinatlas.org/ENSG00000171132-PRKCE/tissue). Therefore, even though ALDH2 is important for analgesia in the spinal cord (British Journal of Anaesthesia 127 (2):296-309, 2021), the PKCε-ALDH2 interaction to mitigate 4HNE-induced pain may happen at peripheral levels. However, further experiments are necessary to investigate this hypothesis.
Is this PCKe-ALDH2 axis specific for in producing analgesia against 4-HNE-induced pain?
Response: This is an interesting question. In the present manuscript we showed that PCKε-ALDH2 axis activation produces analgesia against 4-HNE directly injected in the hind paw and endogenously produced by carrageenan. We showed previously that besides producing 4-HNE, carrageenan also increases malondialdehyde (MDA) and acetaldehyde levels. Of interest, ALDH2 activation with Alda-1 also prevents the increase of these aldehydes (Sci Transl Med 2014;(251):25) and prevents formalin-induced inflammatory pain, suggesting that ALDH2 catalyzes removal of several toxic aldehydes that alter pain responses. Recently, studies demonstrated that ALDH2 activation ameliorates pain in a model of multiple sclerosis by attenuating acrolein, besides MDA and 4-HNE (Neuroscience (458): 31-42, 2021).
Round 2
Reviewer 3 Report
There are a lot to be done to improve
Response: Thank you for the suggestion. We edited the language style and format of the manuscript.